# Phase Angle Is Lower in Older Adults Living with HIV Compared to Geriatric Outpatients: A Case–Control Study

**DOI:** 10.3390/jcm14175941

**Published:** 2025-08-22

**Authors:** Zeynep Şahiner, Merve Güner, Fatma Nisa Balli Turhan, Serdar Ceylan, Arzu Okyar Baş, Merve Hafizoğlu, Didem Karaduman, Cansu Atbaş, Yasemin Polat Özer, Meliha Çağla Sönmezer, Cafer Balci, Burcu Balam Doğu, Mustafa Cankurtaran, Ahmet Çağkan İnkaya, Kutay Demirkan, Serhat Ünal, Meltem Gülhan Halil

**Affiliations:** 1Division of Geriatric Medicine, Department of Internal Medicine, Faculty of Medicine, Hacettepe University, 06230 Ankara, Turkey; mguner@hacettepe.edu.tr (M.G.); serdarceylan@hacettepe.edu.tr (S.C.); arzu0506@hacettepe.edu.tr (A.O.B.); mervehafizogl@hacettepe.edu.tr (M.H.); didemkaraduman@hacettepe.edu.tr (D.K.); cansuatbas@hacettepe.edu.tr (C.A.); yaseminpolat@hacettepe.edu.tr (Y.P.Ö.); caferbalci@hacettepe.edu.tr (C.B.); bbdogu@hacettepe.edu.tr (B.B.D.); mustafacankurtara@hacettepe.edu.tr (M.C.); meltemhalil@hacettepe.edu.tr (M.G.H.); 2Department of Clinical Pharmacy, Faculty of Pharmacy, Gazi University, 06330 Ankara, Turkey; nisa.balli@hacettepe.edu.tr; 3Department of Infectious Diseases, Faculty of Medicine, Hacettepe University, 06080 Ankara, Turkey; melihasonmezer@hacettepe.edu.tr (M.Ç.S.); ahmetcagkaninkaya@hacettepe.edu.tr (A.Ç.İ.); serhatunal@hacetepe.edu.tr (S.Ü.); 4Department of Clinical Pharmacy, Faculty of Pharmacy, Hacettepe University, 06100 Ankara, Turkey; kutaydemirkan@hacettepe.edu.tr

**Keywords:** HIV, geriatric syndromes, phase angle, bioelectrical impedance, people living with HIV (PLWH)

## Abstract

**Background:** Bioelectrical impedance analysis has been used to evaluate phase angle, which predicts cellular health and may even predict survival in people living with HIV. However, the relationship between the phase angle and geriatric syndromes is unclear. This study aims to evaluate geriatric syndromes and how they interact with issues affecting HIV patients by conducting a full geriatric evaluation and comparing phase angles. **Methods:** Fifty people living with HIV and 52 participants without HIV were included in the study. All participants underwent a comprehensive geriatric assessment. BIA was used to determine the phase angle, which was then predicted from impedance measurements. **Results:** The mean age of people living with HIV was 60.0 ± 12.0 years, and that of participants without HIV was 60.0 ± 5.0 years in participants without HIV (*p* = 0.93). The number of drugs used by people living with HIV infection was considerably higher than that used by those in the HIV-negative group (*p* = 0.018). There was a statistically significant difference in the phase angle between without HIV and with HIV. The median [interquartile range (IQR)] phase angle was 7.4 [4.0] degrees, and it was 5.7 [3.2] degrees (*p* = 0.004). **Conclusions:** Phase angle measurements between people living with HIV and without HIV could provide valuable insights into overall health status treatment response and prognosis. Further large-scale research is to corroborate our findings.

## 1. Introduction

Human immunodeficiency virus (HIV) infection is considered a major worldwide health issue affecting approximately 39 million people living with HIV (PLWH), and more than one in every 10 people over the age of 50 years [1]. The availability of present antiviral therapy (ART), particularly since 2000, is credited with the long life expectancy among PLWH [2]. Even though effective ART allows PLWH to live a normal life expectancy comparable to the general population, a key issue remains unsolved to poor transition processes linked with comorbidities and other age-related conditions [3]. As a result, while HIV infection has become a manageable chronic disease, it is still unknown how to best provide geriatric care to PLWH [4] age is an essential risk component of major and even fatal conditions, particularly when combined with HIV.

Early ART initiation ameliorates immune injury, and suppressing the viral load limits life expectancy to those of the uninfected population [5]. People living with HIV experience immune dysregulation and immunosenescence earlier than HIV-negative participants do, including depletion of CD4+ T cells (cluster of differentiation), increased cytokine-secreting senescent CD8+ T cells, and enhanced monocyte stimulation. Functionally, it causes an elevated inflammatory state and a diminished immune response to novel antigens, infections, and vaccinations [6]. Indications for HIV promote the age-related cycle in infected people; the following changes occur at lower ages: increased rates of chronic comorbidities, increased rates of geriatric syndromes, and frailty. Agile immunological alterations increase persistent inflammatory indicators as a result of inflammation [7]. This is why age 50 is usually used as a threshold in the HIV literature to describe aging [8].

The concept behind the introduction of a comprehensive geriatric assessment (CGA) is that a team of health experts can more accurately detect a range of treatable or manageable health issues by conducting an interdisciplinary examination of older adults who are at risk. This method improves medical results and quality of life [9]. Studies have shown that geriatric syndromes are common in PLWH over the age of 50; over half of them have more than two geriatric disorders, and the most common are frailty and neurocognitive disorders [10]. Currently, the CGA approach is rarely used in PLWH because of its association with various morbidities and cognitive impairments. Aged HIV participants’ clinical evaluations need to consider issues of frailty and disability progressively [11].

Geriatric syndromes for PLWH and HIV-negative participants include falls, incontinence, difficulty in activities of daily living, and dependency as a result, sarcopenia, malnutrition, slow gait, and neurocognitive disorders. The incidence of common geriatric disorders is greater in PLWH than in HIV-negative individuals of the same age [12]. This risk is increased by several HIV-specific variables, such as immunological dysregulation, chronic inflammation, long-term ART toxicity, and sociobehavioral problems [13] Importantly, aging is also a significant risk factor for serious comorbidities, such as HIV, which can be potentially fatal.

The phase angle (PhA), a metric developed from bioimpedance analysis (BIA), represents the ratio of reactance (Xc) to resistance. BIA estimates body composition via a low-frequency electric current; Xc represents the membrane’s ability to hold an electrical potential, and R represents the capacitance of body fluids and electrolytes to electric current. As a result, PhA provides information on tissue hydration status, cell membrane mass, cellular health, membrane integrity, and function. Lower phase angle values have been associated with decreased cell integrity and cell death [14].

Over the last decade, there has been an increase in research into the use of raw bioimpedance measurements such as resistance (R), reactance (Xc), or phase angleangles to calculate body composition, and there is ample evidence that a reduced bioimpedance phase angle is associated with a variety of clinical outcomes, including mortality in critically ill patients, patients with kidney disease, heart disease, and patients with cancer [15]. PhA has also been linked to several symptoms of aging, including sarcopenia, frailty, malnutrition, and falls [16].

However, the integration of PhA with CGAwhich evaluates physical, cognitive, nutritional, and functional domains remains rare in research on older adults living with HIV. To address this gap, our case–control study compares PhA and CGA-derived geriatric syndromes in PLWH aged ≥ 50 against age-matched HIV-negative geriatric outpatients. We hypothesize that PLWH will demonstrate lower PhA values and a higher burden of geriatric syndromes, reflecting accelerated cellular aging and functional decline.

## 2. Materials and Methods

### 2.1. Study Population

A total of 102 participants, 50 of whom were PLWH and 52 of whom were without HIV infection, who were admitted to the infectious diseases and ambulatory clinic for geriatrics from December 2022 until August 2023 were included in this prospective study. The CGA was applied to all participants. Demographic data and anthropometric measurements, such as age, sex, education, marital status, place of residence, and cohabitation, were recorded, and geriatric syndromes, including frailty, number of medications, polypharmacy, malnutrition, cognitive function, and sarcopenia, were recorded for each person. The control group was matched with the case group from the same source in terms of similar age, socioeconomic level, and comorbid diseases.

The inclusion criteria for PLWH over 50 years of age were having completed the consent form, being open to communication, being on antiretroviral therapy, and having a negative viral load for at least three months. The exclusion criteria were participants’ request to be excluded from the study and failure to fill out the consent form. The inclusion criteria for the case group were being over 50 years of age and being HIV negative.

### 2.2. Comprehensive Geriatric Assesment

#### 2.2.1. Physical Function Assessment

The Lawton-Brody Instrumental Activities of Daily Living (IADL) and Katz Activities of Daily Living (ADL) were used to measure individuals’ functional abilities. Katz’s ADL test is evaluated on a scale of more than 6 points by asking how independently people perform basic care and activities related to daily living, and the score increases as independence increases [17]. The IADL measures the ability of participants to perform complex daily activities, and its score is calculated as over 8 points [18].

#### 2.2.2. Malnutrition Assessment

The nutritional status of the participants was examined via the Mini Nutritional Assessment Short Form (MNA-SF). During the MNA-SF test, people’s BMI, weight loss in the last 3 months, presence of psychosocial stress or an acute illness in the last 3 months, mobility problems, neurocognitive impairment, and appetite were assessed. Normal scores were defined as those with scores greater than 11 points, malnutrition risk was defined as those with scores ranging from 8–11 points, and malnutrition risk was defined as those with scores ≤ 7 points [19]. Polypharmacy was defined as the use of more than five drugs [20].

#### 2.2.3. Frailty Assessment

Frailty was assessed with three different tools. The Clinical Frailty Scale (CFS) score ranged from 1 to 9. CFS was performed by the same experienced physician according to the physician’s clinical opinion and to assess frailty status. According to the accepted definitions, participants were divided into two groups: nonfrail/robust (CFS < 4) and living with frailty (CFS  ≥  4) [21,22]. The modified FRIED frailty index (FFI), which takes into account weight loss, fatigue, reduced energy expenditure, decreased strength, and walking speed. Each item was assessed as 0 or 1, and a total score of 0 was considered robust, 1–2 prefrail, and 3 or above frail [23,24]. The FRAIL scale is a practical tool that helps determine frailty by evaluating individuals in terms of fatigue, resistance, walking, disease, and weight loss. If the total score given by individuals to the five questions on the FRAIL scale is 0 (zero), they are considered nonfrail/robust; if the total score is 1–2 points, they are considered prefrail; and if the total score above 2, they are considered frail [25,26].

#### 2.2.4. Cognitive Assessment

Quick mild cognitive impairment-Turkish (Qmc-TR) test was performed to evaluate cognitive function. In the qmci-TR, which is superior in distinguishing MCI, different cutoff values are used for different education levels. In the Turkish cutoff determination study according to different age and education status groups [27,28].

#### 2.2.5. Sarcopenia and Body Composition Assessment

Muscle strength was assessed via a dynamometer for Takei grip strength. The European Working Group on Sarcopenia in Older People (EWSGOP) updated sarcopenia criteria were used to determine the cutoff values, and handgrip strength (HGS) < 16 kg and < 27 kg were used to indicate low muscle strength in men and women, respectively [29]. Muscle mass was measured by BIA and ultrasound (US). BIA was performed with a Bodystat Quadscan 4000 device. Fat-free mass (FFM) was assessed via BIA, and skeletal muscle mass (SMM) was determined via the following validated equation: SMM (kg) = FFM ∗ 0.566. The skeletal muscle index (SMI) (SMM divided by height squared) was used to estimate muscle mass, and the phase angle (PhA) was assessed for each participant. Muscle US was measured with a 5 cm linear probe (LOGIQ 200 PRO, General Electric Medical Systems) at 10 to 12 MHz. All US parameters were collected by the same experienced physician. The rectus abdominis (RA), external abdominal oblique (EO), internal abdominal oblique (IO), transversus abdominalis (TA), gastrocnemius medial (GM), and rectus femoris (RF) muscles were tested toward the end of exhalation to reduce the influence of breathing.

### 2.3. Statistical Analyses

SPSS software, version 23, was used to conduct the statistical analyses. To ascertain whether the variables were normally distributed, they were evaluated analytically (Kolmogorov–Smirnov test) and visually (histograms, probability plots). The means ± standard deviations (SDs) for normally distributed data, medians [IQRs] for nonnormally distributed variables, and percentages for categorical variables were used for descriptive analyses. Differences in categorical variables were compared via the chi-square test. Continuous variables were compared via the Mann–Whitney U test. The associations between phase angle and continuous clinical variables (e.g., age, handgrip strength, skeletal muscle index, MNA-SF score, CFS score, number of medications, Qmci score) were evaluated using Spearman’s rank correlation analysis. To identify independent predictors of phase angle, multivariable linear regression analysis was performed. Variables with clinical relevance or those with a *p* < 0.10 in univariate analyses were included in the model. The assumptions of linearity, homoscedasticity, and multicollinearity were checked prior to model inclusion. Additionally, the discriminatory ability of phase angle to distinguish between HIV positive and HIV negative individuals was assessed using Receiver Operating Characteristic (ROC) curve analysis. The area under the curve (AUC) was calculated, and optimal cut-off points were identified based on the Youden index. Sensitivity and specificity were also reported for clinically relevant cut-off values. We evaluated discrimination using two multivariable logistic models: a base model (age, sex, BMI, SMI, maximal handgrip strength, FRAIL score, CFS, MNA-SF) and the same model plus phase angle (PhA). Continuous predictors were z-standardized and sex was modeled as categorical; analyses used complete cases. Discrimination was summarized by ROC AUC with 95% stratified bootstrap confidence intervals. and the incremental value of PhA was tested via the paired AUC difference (ΔAUC) with a bootstrap two-sided *p*-value. For clinical interpretability, Youden-optimal thresholds were used to derive confusion/error matrices and operating metrics A two-sided *p*-value < 0.05 was considered statistically significant.

## 3. Results

The median age of the PLWH was 60.0 [12.0] years, whereas the median age of the participants without HIV infection was 60.0 [5.0] years, and the change was not considerable (*p* = 0.93). The number of male PLWH was 37 (74.0%). Other demographic features and anthropometric measurements were similar between the two groups, except for marital status (*p* = 0.01) (Table 1).

Table 2 shows the characteristics of the CGAs of the study population according to HIV infection status. For PLWH without HIV, the median Katz ADL score was 6.0 [0.0] (*p* = 0.20). The percentage of participants living with frailty evaluated with the CFS was highergreater in the PLWH than in those without HIV (18% vs. 13.5%, respectively); however, the difference was not significant (*p* = 0.53). According to the FFI, the ratios of frailty were 20% and 19.2% in PLWH and those without HIV infection, respectively (*p* = 0.92), and according to the FRAIL scale, 24% of the PLWH and 17.3% of the participants without HIV infection were living with frailty (*p* = 0.40). There was no significant difference in malnutrition, HGS, or gait speed between PLWH and participants without HIV infection. No differences were observed in the cognitive evaluation between the two groups (*p* = 0.53).

The muscle assessment results are shown in Table 3 no differences were found between the PLWH and participants without HIV infection according to muscle US or BIA. However, PhA was significantly lower in PLWH than in participants without HIV infection. It was 5.7 [3.2] degrees in PLWH, and it was 7.4 [4.0] degrees in participants without HIV infection (*p* = 0.004).

Spearman correlation analysis revealed that phase angle was positively correlated with skeletal muscle index (SMI) (r = 0.42, *p* < 0.001) and handgrip strength (r = 0.38, *p* < 0.001) (Appendix A).

In the base multivariable logistic model without PhA, none of the candidate predictors reached conventional statistical significance for the endpoint. Discrimination was modest (AUC ≈ 0.687, 95% CI 0.639–0.828). At the Youden-optimal threshold, sensitivity was ~50% and specificity ~83%, with accuracy ~65% (Figure 1, Table 4 and Table 5).

After adding PhA to the same covariates, no predictor including PhA was independently associated with the endpoint in the multivariable model (PhA OR ≈ 0.964, 95% CI 0.865–1.074, *p* ≈ 0.50). Model discrimination remained modest (AUC ≈ 0.695, 95% CI 0.653–0.839), with no meaningful improvement versus the base model (ΔAUC ≈ 0.008, *p* ≈ 0.467). At the Youden-optimal threshold, sensitivity was ~48% and specificity ~85%, with accuracy ~65% (Table 4 and Table 6).

Appendix A shows receiver operating characteristic (ROC) analysis was conducted to evaluate the discriminatory capacity of phase angle in identifying HIV-positive individuals. At a cut-off value of 5.0 degrees, the sensitivity and specificity were 15.1% and 73.3%, respectively, with an area under the curve (AUC) of 0.69 (95% CI: 0.58–0.80) and a statistically significant *p*-value (*p* = 0.004). Although the discriminatory power was moderate, the relatively low sensitivity at this cut-off suggests limited utility of phase angle as a standalone diagnostic marker. However, its moderate specificity may support its use as part of a broader clinical assessment, particularly in identifying individuals at higher risk of cellular or functional decline.

Additionally characteristic features of HIV positive patients are also given in Appendix A.

## 4. Discussion

Our study aimed to explore the clinical and functional determinants of phase angle in older adults with and without HIV, with particular attention to body composition, nutritional status, and frailty indicators. Consistent with previous research, we observed that individuals living with HIV had lower phase angle values compared to HIV-negative controls [30,31]. This finding aligns with earlier reports suggesting that chronic HIV infection, even in the era of effective antiretroviral therapy (ART), is associated with persistent changes in cellular membrane integrity, increased inflammation, and loss of muscle mass, all of which contribute to a lower phase angle [7,12].

However, when adjusted for age, muscle mass, and other relevant covariates in a multivariable model, HIV status was no longer an independent predictor of phase angle. This contrasts with earlier studies that reported a direct association between HIV and lower phase angle [16,30], and suggests that in clinically stable, well-managed individuals, other factors such as aging and sarcopenia may exert a more dominant effect on cell membrane function than HIV itself. The absence of a significant HIV effect in our adjusted model could be attributed to the older age and preserved virological status of our cohort, as well as possible survivor bias.

The strongest independent associations with phase angle were observed for age and skeletal muscle index (SMI), highlighting the influence of biological aging and muscle mass on cellular health. These findings are supported by prior studies reporting that phase angle is a surrogate marker of muscle integrity, and declines progressively with aging and sarcopenia [29,30,31,32,33]. Moreover, significant correlations between phase angle and handgrip strength reinforce the idea that phase angle reflects not only structural but also functional aspects of the neuromuscular system [34].

Interestingly, our ROC analysis showed that phase angle had only moderate capacity to distinguish between HIV-positive and HIV-negative individuals. This finding contrasts with earlier reports that suggested phase angle might serve as a useful diagnostic or prognostic biomarker in HIV [14,31]. The lower area under the curve (AUC) and modest sensitivity/specificity in our sample may be explained by the relatively small sample size, the older age of participants, and the inclusion of clinically stable patients with well-controlled HIV infection. It is also possible that phase angle is more sensitive to general health decline, such as frailty and malnutrition, rather than HIV status.

Overall, our findings suggest that phase angle in older adults particularly those with HIV reflects a complex interplay between aging, muscle mass, functional reserve, and comorbidity, rather than a direct effect of the HIV infection itself. This underscores the need to interpret phase angle in the context of a multidimensional geriatric framework. Our results advocate for the routine use of phase angle in comprehensive geriatric assessments, especially for identifying sarcopenia, nutritional risk, and vulnerability to adverse outcomes, regardless of HIV status.

The limitations of our study should be noted. The participants included were selected from a referral center; therefore, they may not be typical of all individuals with HIV. In addition, this study is cross-sectional and precludes the possibility of recognizing any cause and effect relationships between BIA-derived measures and clinical, lifestyle, and dietary cofactors. This is among the first studies to evaluate the relationships between BIA-derived measures such as the FFMI and PhA and cofactors such as socioeconomic, medical, lifestyle, nutritional, biochemical, and virological variables in patients who were chronically infected with HIV and those without.

## 5. Conclusions

This study demonstrates that integrating PA measurement within the CGA framework can enhance early identification of older adults living with HIV who are at risk of frailty, low muscle mass, and functional decline. By highlighting the roles of aging and skeletal muscle index in determining phase angle, our findings support the use of these assessments to guide clinical interventions aimed at preserving independence and physiological reserve in this population. Ultimately, routine evaluation of phase angle alongside CGA may help develop tailored strategies to sustain the health and well being of aging individuals with HIV.

## Figures and Tables

**Figure 1 jcm-14-05941-f001:**
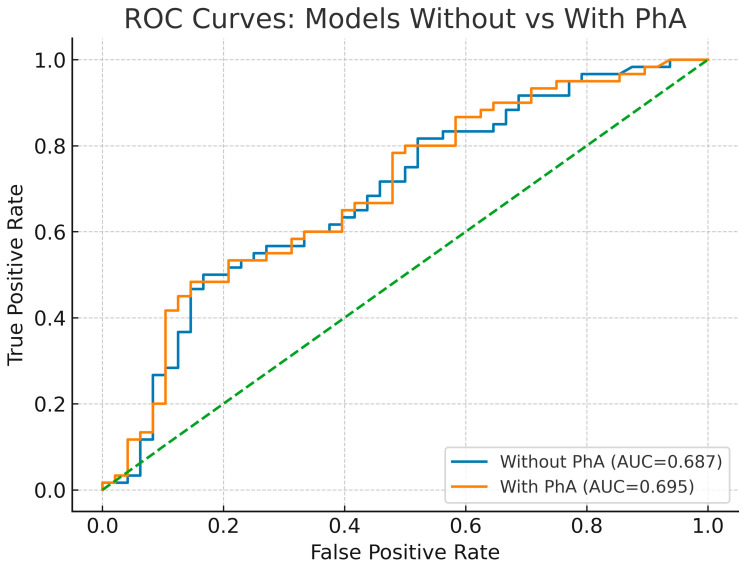
ROC curves for models with vs. without PhA. PhA, phase angle; ROC, receiver operating characteristic.

**Table 1 jcm-14-05941-t001:** Comparison of the study population according to HIV infection status.

	HIV Positive (*n* = 50)	HIV Negative (*n* = 52)	*p*
Age, years, median (IQR)	60.0 [12.0]	60.0 [5.0]	0.93
Sex, male, *n* (%)	37 (74.0)	36 (69.2)	0.59
Marital status, married, *n* (%)	32 (64.0)	44 (84.6)	0.017
Education, <8 years, *n* (%)	18 (36.0)	14 (26.9)	0.32
MUAC, cm, mean ± SD	30.7 ± 3.2	30.1 ± 3.8	0.46
CC, cm, median (IQR)	36.0 [5.0]	35.0 [6.0]	0.49
WC, cm, mean ± SD	103.0 ± 18.9	99.1 ± 10.8	0.78
HC, cm, mean ± SD	108.1 ± 8.3	105.6 ± 9.8	0.16
BMI, kg/m^2^, mean ± SD	28.5 ± 5.6	27.9 ± 4.4	0.62

Variables are presented as *n* (%), mean ± SD or median [IQR]. BM, Body mass index; CC, Calf circumference; Cm, Centimeter; HC, Hip circumference; MUAC, Mid upper arm circumference; kg, Kilogram; m^2^, Square meters; WC, Waist circumference.

**Table 2 jcm-14-05941-t002:** The features of comprehensive geriatric assessments of the study population according to HIV infection status.

	HIV Positive (*n* = 50)	HIV Negative (*n* = 52)	*p*
Katz Basic ADL scores, median (IQR)	6.0 [0.0]	6.0 [0.0]	0.20
Lawton Brody IADL score, median (IQR)	8.0 [0.0]	8.0 [0.0]	0.28
Living with frailty, CFS, *n* (%)	9 (18.0)	7 (13.5)	0.53
CFS Score, median (IQR)	3.0 [1.0]	2.0 [1.0]	0.11
Living with frailty, FFI, *n* (%)	10 (20.0)	10 (19.2)	0.92
FFI Score, median (IQR)	0.0 [0.0]	0.0 [0.0]	0.90
Living with frailty, FRAIL, *n* (%)	12 (24.0)	9 (17.3)	0.40
FRAIL score, median (IQR)	0.0 [0.0]	0.0 [0.0]	0.43
Malnutrition, *n* (%)	3 (6.0)	4 (7.7)	1.0
MNA-SF score, median (IQR)	14.0 [0.0]	14.0 [1.0]	0.59
HGS, kg, mean ± SD	32.7 ± 8.4	31.3 ± 9.0	0.93
Probable Sarcopenia, *n* (%)	5 (10.0)	6 (11.5)	0.80
4-meter walking, sec, median (IQR)	3.3 [1.3]	3.7 [1.1]	0.94
Gait speed, <0.8 m/sec, *n* (%)	5 (10.0)	5 (9.6)	0.95
Polypharmacy, *n* (%)	12 (24.0)	6 (11.5)	0.099
Drug number, median (IQR)	3.0 [2.0]	2.0 [3.0]	0.018
Qmcı, score, median (IQR)	84.0 [15.5]	81.0 [17.0]	0.53

Variables are presented as *n* (%), mean ± SD or median [IQR]. ADL, Activities of daily living; CFS, Clinical Frailty Scale; FFI, Fried frailty index; HGS, Handgrip Strength; IADL, Instrumental activities of daily living; MNA-SF, Mini Nutritional Assessment-Short Form; QMCI, quick mild cognitive impairment.

**Table 3 jcm-14-05941-t003:** Muscle assessments of the study population according to HIV infection status.

	HIV Positive (*n* = 50)	HIV Negative (*n* = 52)	*p*
Gastrocnemius Medialis MT., mm, mean ± SD	16.07 ± 3.8	16.04 ± 3.2	0.57
Rectus Femoris MT., mm, mean ± SD	16.75 ± 3.89	16.68 ± 4.3	0.82
Rectus Femoris CSA, mm, mean ± SD	7.85 ± 2.50	7.65 ± 2.75	0.67
Rectus Abdominis MT., mm, mean ± SD	9.38 ± 1.90	8.87 ± 2.04	0.080
External Oblique MT., mm, mean ± SD	4.40 ± 1.21	4.63 ± 1.41	0.37
Internal Oblique MT., mm, mean ± SD	6.36 ± 1.68	6.53 ± 2.20	0.40
Transverse Abdominis MT., mm, mean ± SD	4.57 ± 1.56	4.66 ± 1.46	0.81
FFMI, kg/m^2^, median (IQR)	18.9 [3.3]	19.5 [5.1]	0.53
SMI, kg/m^2^, mean ± SD	9.84 ± 4.11	9.03 ± 3.66	0.12
Phase Angle, median (IQR)	5.7 [3.2]	7.4 [4.0]	0.004

Variables are presented as *n* (%), mean ± SD or median [IQR]. FFMI, Fat-Free Mass Index; kg, Kilogram; MT, Muscle thickness; mm, Millimeter; m^2^, Square meters; SMI, Skeletal muscle mass index.

**Table 4 jcm-14-05941-t004:** ROC AUCs for models with vs. without PhA.

Model	AUC (95% CI)	*p*-Value
Without PhA	0.687 (0.639–0.828)	<0.001
With PhA	0.695 (0.653–0.839)	<0.001
Difference (With − Without)	0.008 (0.008–0.071)	0.467

PhA, phase angle; ROC, receiver operating characteristic; AUC, area under the curve; ΔAUC, AUC difference (with − without PhA); CI, confidence interval; *p*-value.

**Table 5 jcm-14-05941-t005:** Logistic Regression Model A (without PhA): ORs, 95% CIs, *p*-values.

Predictor	OR	95% CI Lower	95% CI Upper	*p*-Value
Age	0.975	0.904	1.051	0.505
Sex	0.29	0.053	1.601	0.156
Bmı	1.109	0.924	1.331	0.268
SMI	0.838	0.629	1.116	0.226
Handgrip	0.981	0.913	1.054	0.604
Frail	1.254	0.476	3.306	0.647
Cfs	0.559	0.262	1.192	0.132
Mna_sf	0.759	0.532	1.083	0.128

OR, odds ratio; CI, confidence interval; BMI, body mass index; SMI, skeletal muscle index; HGS, handgrip strength (maximal); FRAIL, FRAIL score; CFS, Clinical Frailty Scale; MNA-SF, Mini Nutritional Assessment–Short Form; *p*-value.

**Table 6 jcm-14-05941-t006:** Logistic Regression Model B (with PhA): ORs, 95% CIs, *p*-values.

Predictor	OR	95% CI Lower	95% CI Upper	*p*-Value
Age	0.969	0.897	1.047	0.432
Sex	0.314	0.057	1.738	0.184
Bmı	1.096	0.912	1.318	0.327
SMI	0.856	0.640	1.143	0.292
Handgrip	0.978	0.909	1.052	0.545
Frail	1.171	0.435	3.152	0.675
Cfs	0.572	0.268	1.221	0.148
Mna_sf	0.751	0.522	1.080	0.122
PhA	0.964	0.865	1.074	0.504

OR, odds ratio; CI, confidence interval; PhA, phase angle; BMI, body mass index; SMI, skeletal muscle index; HGS, handgrip strength (maximal); FRAIL, FRAIL score; CFS, Clinical Frailty Scale; MNA-SF, Mini Nutritional Assessment–Short Form; *p*-value.

## Data Availability

The datasets used and/or analyzed in the current work will be made available by the corresponding author upon reasonable request.

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
