# Peer review of "Phase Angle Is Lower in Older Adults Living with HIV Compared to Geriatric Outpatients: A Case–Control Study"

_jcm, 2025, doi:10.3390/jcm14175941_

Round 1
Reviewer 1 Report
Comments and Suggestions for Authors
The manuscript "Phase Angle Is Lower in Older Adults Living with HIV Compared to Geriatric Outpatients: A Case–Control Study" is clear and well-structured. The manuscript addresses a relevant and up to date topic in geriatric& HIV medicine. The integration of phase angle (PhA) measurements within a comprehensive geriatric assessment (CGA) framework is a combination which contributrs to an important practical clinical perspective.
The manuscript is logically organized and clearly written. The objectives, methodology, results, and conclusions are understandable. The manuscript contributets to understanding aging in people living with HIV (PLWH), particularly the functional and nutritional dimensions.
The majority of the references are recent (within the last 5 years) and relevant, with a broad spectrum perspectives.
The study is scientifically sound with appropriate methodology for a case–control design. CGA tools are correctly used, and the use of BIA for PhA measurement is goes along with current practice. Ghe description of the control group matching was adequate.
The methods section is detailed. clear The definitions of assessment tools and statistical methods are clear so the methodology is reproducible. Validated scales were used (CFS, FRAIL, MNA-SF, Qmci).
Tables are comprehensive, well-labeled, and so the data are clearly presented.
The statistical analyses, including regression and ROC analysis, are appropriate and well interpreted. I find that the predictive value of PhA in ROC analysis and multivariable modeling are modest , so maybe a professional statitician should be consulted. But I do not find this as a reason not to accept the manuscript.
The conclusions are consistent with the results . The limitations are presented appropriately, such as sample size and cross-sectional design.
The practical clinical implications are clearly stated.
This is a well-conducted and clinically important study.
***Addition:
1.The main question addressed by the study is whether older adults living with HIV (PLWH) exhibit lower phase angle (PhA) values, which is an indicator of cellular health—compared to age-matched, HIV-negative geriatric outpatients. Another topic is whether this difference correlates with comprehensive geriatric assessment (CGA) parameters such as frailty, malnutrition, sarcopenia, and cognitive decline.
2.The topic is both original and highly relevant because it analyzes a well-defined gap in the literature by integrating phase angle measurements with a multidimensional geriatric assessment in older PLWH—a population often underrepresented in geriatric research. It is known that PhA has been widely studied in critically ill or hospitalized patients, and that its specific utility in aging PLWH as part of a CGA framework has not been investigated in details. So this manuscript provides good analysis of aging with HIV in the context of functional health and cellular integrity.
3.This work demonstrates a statistically significant reduction in PhA among PLWH compared to HIV-negative peers.
Correlating PhA with multiple geriatric variables (e.g., SMI, handgrip strength, number of medications, frailty scores).
It accentuaze the limited standalone diagnostic utility of PhA for HIV status, but underscoring its value as part of an integrated assessment for frailty and sarcopenia.
It suppirts the routine use of PhA in geriatric HIV care to guide individualized clinical interventions.
4. Possible improvement:
Sample size: Although statistically justified, a larger and more diverse sample would improve statistical power in multivariable analyses.
The cohort is predominantly male;
5.The conclusions are consistent with the presented data and adequately address the main question.
The authors appropriately acknowledge the limitations of their cross-sectional design, the modest discriminatory power of PhA for HIV status, and the influence of aging and muscle mass on PhA. The call for incorporating PhA into routine CGA in aging PLWH is well justified by the observed correlations and the theoretical framework discussed.
6.The references are up-to-date, relevant, and understandable. The authors have cited foundational literature on PhA, sarcopenia, frailty, and HIV aging, including both international studies and regionally validated scales.
7.Tables are clearly formatted, comprehensive, and include relevant statistical metrics. (Multivariable regression, ROC analysis).
Figures were not included in the draft.
Author Response
Phase Angle Is Lower in Older Adults Living with HIV Compared to Geriatric Outpatients: A Case–Control Study
Response to Reviewer 1 Comments
|
||
1. Summary |
|
|
Thank you very much for taking the time to review this manuscript. Please find the detailed responses below and the corresponding revisions/corrections highlighted/in track changes in the re-submitted files.
|
||
2. Questions for General Evaluation |
Reviewer’s Evaluation |
Response and Revisions |
Does the introduction provide sufficient background and include all relevant references? |
Yes |
We are grateful for your valuable comment. |
Are all the cited references relevant to the research? |
Yes |
We are grateful for your valuable comment. |
Is the research design appropriate? |
Can be improved |
We are grateful for your valuable comment. |
Are the methods adequately described? |
Yes |
We are grateful for your valuable comment. |
Are the results clearly presented? |
Yes |
We are grateful for your valuable comment. |
Are the conclusions supported by the results? |
Yes |
We are grateful for your valuable comment. |
3. Point-by-point response to Comments and Suggestions for Authors |
||
Comments 1: The manuscript "Phase Angle Is Lower in Older Adults Living with HIV Compared to Geriatric Outpatients: A Case–Control Study" is clear and well-structured. The manuscript addresses a relevant and up to date topic in geriatric& HIV medicine. The integration of phase angle (PhA) measurements within a comprehensive geriatric assessment (CGA) framework is a combination which contributrs to an important practical clinical perspective.
|
||
Response 1: We are honored by your very valuable comment. |
||
Comments 2: The manuscript is logically organized and clearly written. The objectives, methodology, results, and conclusions are understandable. The manuscript contributets to understanding aging in people living with HIV (PLWH), particularly the functional and nutritional dimensions. |
||
Response 2: Your valuable comments are very important to us.
|
||
Comments 3: The majority of the references are recent (within the last 5 years) and relevant, with a broad spectrum perspectives. |
||
Response 3: Dear reviewer, rest assured that we worked hard to make it like this. |
||
Comments 4: The study is scientifically sound with appropriate methodology for a case–control design. CGA tools are correctly used, and the use of BIA for PhA measurement is goes along with current practice. Ghe description of the control group matching was adequate. |
||
Response 4: We are grateful for your valuable comment. Yes, we made a lot of effort to collect this patient group and create an equivalent control group. |
||
Comments 5: The methods section is detailed. clear The definitions of assessment tools and statistical methods are clear so the methodology is reproducible. Validated scales were used (CFS, FRAIL, MNA-SF, Qmci). |
||
Response 5: Thank you for your positive and encouraging feedback regarding the methodology. |
||
Comments 6: Tables are comprehensive, well-labeled, and so the data are clearly presented. |
||
Response 6: Thank you for your thoughtful evaluation of our tables and statistical analyses. |
||
Comments 7: The conclusions are consistent with the results . The limitations are presented appropriately, such as sample size and cross-sectional design.
|
||
Response 7: Thank you for your positive and encouraging feedback. |
||
Comments 8: ***Addition: 2.The topic is both original and highly relevant because it analyzes a well-defined gap in the literature by integrating phase angle measurements with a multidimensional geriatric assessment in older PLWH—a population often underrepresented in geriatric research. It is known that PhA has been widely studied in critically ill or hospitalized patients, and that its specific utility in aging PLWH as part of a CGA framework has not been investigated in details. So this manuscript provides good analysis of aging with HIV in the context of functional health and cellular integrity. 3.This work demonstrates a statistically significant reduction in PhA among PLWH compared to HIV-negative peers. Correlating PhA with multiple geriatric variables (e.g., SMI, handgrip strength, number of medications, frailty scores). It accentuaze the limited standalone diagnostic utility of PhA for HIV status, but underscoring its value as part of an integrated assessment for frailty and sarcopenia. It suppirts the routine use of PhA in geriatric HIV care to guide individualized clinical interventions.
|
||
Response 8: Thank you for your comprehensive and insightful summary of our work. |
||
Comments 9: Possible improvement: Sample size: Although statistically justified, a larger and more diverse sample would improve statistical power in multivariable analyses. The cohort is predominantly male; |
||
Response 9: Thank you for this valuable suggestion. We acknowledge that, although the sample size was statistically justified, a larger and more diverse cohort would indeed increase the statistical power of multivariable analyses and potentially enhance the generalizability of the findings. We also recognize the gender imbalance in our study population, which reflects the demographic characteristics of the HIV-positive population in our clinical setting. We have noted these points in the limitations section and will aim to address them in future research through broader recruitment strategies. |
||
Comments 10: 5.The conclusions are consistent with the presented data and adequately address the main question. 6.The references are up-to-date, relevant, and understandable. The authors have cited foundational literature on PhA, sarcopenia, frailty, and HIV aging, including both international studies and regionally validated scales. 7.Tables are clearly formatted, comprehensive, and include relevant statistical metrics. (Multivariable regression, ROC analysis). Figures were not included in the draft. |
||
Response 10: Thank you for your constructive and encouraging feedback. |
Reviewer 2 Report
Comments and Suggestions for Authors
The goal of the paper is to access whether the results
of bioelectric impedance analysis (PhA) could be the valid factor in the survival
studies for PLWH. For this goal a sample of 50 PLWH and a control sample of 52
geriatric participants was studied. The associtions between phase angle and
continuous clinical variables were evaluated using Spearman's rank correlation
analysis. This test is distribution-free, so the discussion about the Gaussion
character of the clinical variales does not make much sense. Receiver Operation
Charactetistic (ROC) is analysed. It would make sense to compare the Areas Under
the Curve (AUC) for survival predictions based on the characteristic including/excluding
PhA. The statistical analysis should be presented in more details and illustrated by
the proper figures. Note that the linear regression presented in Table 4 does not
make much sense in this context. It would make sense to present the confusion/error matrices
for the predictions including/excluding PhA in the tradition of AI literature.
Author Response
For review article
Response to Reviewer 2 Comments
|
||
1. Summary |
|
|
Thank you very much for taking the time to review this manuscript. Please find the detailed responses below and the corresponding revisions/corrections highlighted/in track changes in the re-submitted files.
|
||
2. Questions for General Evaluation |
Reviewer’s Evaluation |
Response and Revisions |
Does the introduction provide sufficient background and include all relevant references? |
Can be improved |
We are grateful for your valuable comment. |
Are all the cited references relevant to the research? |
Can be improved |
We are grateful for your valuable comment. |
Is the research design appropriate? |
Can be improved |
We are grateful for your valuable comment. |
Are the methods adequately described? |
Can be improved |
We are grateful for your valuable comment. |
Are the results clearly presented? |
Can be improved |
We are grateful for your valuable comment. |
Are the conclusions supported by the results? |
Must be improved |
We are grateful for your valuable comment. |
Comments 1: The goal of the paper is to access whether the results of bioelectric impedance analysis (PhA) could be the valid factor in the survival studies for PLWH. For this goal a sample of 50 PLWH and a control sample of 52 geriatric participants was studied.
|
||
Response 1: Thank you for summarizing the main aim of our study. We appreciate your clear restatement of our objective, which is to evaluate whether phase angle (PhA) derived from bioelectrical impedance analysis could serve as a valid factor in survival-related assessments for PLWH. As you noted, our study compared of 50 PLWH with a control group of 52 geriatric participants, aiming to explore not only group differences but also the potential clinical implications of PhA in this specific population. |
||
Comments 2: The associtions between phase angle and continuous clinical variables were evaluated using Spearman's rank correlation analysis. This test is distribution-free, so the discussion about the Gaussion character of the clinical variales does not make much sense. |
||
Response 2: Thank you for the clarification regarding the statistical approach. Because we used Spearman’s rank correlation a distribution-free method we removed the extended discussion of Gaussianity and streamlined the text. We keep a brief summary in the main Results, and provide the full correlation table in the Supplement. |
||
Comments 3: Receiver Operation Charactetistic (ROC) is analysed. It would make sense to compare the Areas Under the Curve (AUC) for survival predictions based on the characteristic including/excluding PhA. |
||
Response 3: Thank you for this helpful suggestion. Our dataset does not include time-to-event outcomes; therefore, instead of survival-specific AUC, we implemented your recommendation by comparing the discriminative performance of two multivariable logistic models without vs with PhA. We now report ROC AUCs with 95% stratified bootstrap CIs (B=2000) and a paired test of ΔAUC, and provide the corresponding ROC curves and confusion/error matrices at the Youden-optimal threshold (see Table 4,5,6 and Figure). In a future prospective cohort with survival data, we will extend this analysis using time-dependent AUC/C-index. |
||
Comments 4: The statistical analysis should be presented in more details and illustrated by the proper figures. Note that the linear regression presented in Table 4 does not make much sense in this context. It would make sense to present the confusion/error matrices for the predictions including/excluding PhA in the tradition of AI literature. |
||
Response 4: We appreciate the suggestion. We have removed the multivariable linear regression from the main text. Logistic regression was performed instead of linear (Tables 5,6) These changes do not affect the study’s conclusions; the Statistical Analysis section has been updated accordingly. |
Round 2
Reviewer 2 Report
Comments and Suggestions for Authors
The revision is adequate.